# Performance Investigations of InAs/InP Quantum-Dash Semiconductor Optical Amplifiers with Different Numbers of Dash Layers

**DOI:** 10.3390/mi14122230

**Published:** 2023-12-12

**Authors:** Youxin Mao, Xiaoran Xie, Chunying Song, Zhenguo Lu, Philip J. Poole, Jiaren Liu, Mia Toreja, Yang Qi, Guocheng Liu, Pedro Barrios, Daniel Poitras, John Weber, Ping Zhao, Martin Vachon, Mohamed Rahim, Penghui Ma, Silas Chen, Ahmad Atieh

**Affiliations:** 1Advanced Electronics and Photonics Research Centre, National Research Council, Ottawa, ON K1A 0R6, Canada; xiaoran.xie@mail.concordia.ca (X.X.); chun-ying.song@nrc-cnrc.gc.ca (C.S.); zhenguo.lu@nrc-cnrc.gc.ca (Z.L.); philip.poole@nrc-cnrc.gc.ca (P.J.P.); jiaren.liu@nrc-cnrc.gc.ca (J.L.); toreja.mia@gmail.com (M.T.); yang.qi@nrc-cnrc.gc.ca (Y.Q.); guocheng.liu@nrc-cnrc.gc.ca (G.L.); pedro.barrios@nrc-cnrc.gc.ca (P.B.); daniel.poitras@nrc-cnrc.gc.ca (D.P.); ping.zhao@nrc-cnrc.gc.ca (P.Z.); martin.vachon@nrc-cnrc.gc.ca (M.V.); mohamed.rahim@nrc-cnrc.gc.ca (M.R.); penghui.ma@nrc-cnrc.gc.ca (P.M.);; 2Optiwave Systems Inc., 7 Capella Court, Suite 300, Ottawa, ON K1N 6N5, Canada; ahmad.atieh@optiwave.com

**Keywords:** semiconductor optical amplifier, InAs/InP quantum dot/dash, chip gain, 3 dB saturated output power, noise figure

## Abstract

We present here a performance comparison of quantum-dash (Qdash) semiconductor amplifiers (SOAs) with three, five, eight, and twelve InAs dash layers grown on InP substrates. Other than the number of Qdash layers, the structures were identical. The eight-layer Qdash SOA gave the highest amplified spontaneous emission power (4.3 dBm) and chip gain (26.4 dB) at 1550 nm, with a 300 mA CW bias current and at 25 °C temperature, while SOAs with fewer Qdash layers (for example, three-layer Qdash SOA), had a wider ASE bandwidth (90 nm) and larger 3 dB gain saturated output power (18.2 dBm) in a shorter wavelength range. The noise figure (NF) of the SOAs increased nearly linearly with the number of Qdash layers. The longest gain peak wavelength of 1570 nm was observed for the 12-layer Qdash SOA. The most balanced performance was obtained with a five-layer Qdash SOA, with a 25.4 dB small-signal chip gain, 15.2 dBm 3 dB output saturated power, and 5.7 dB NF at 1532 nm, 300 mA and 25 °C. These results are better than those of quantum well SOAs reported in a recent review paper. The high performance of InAs/InP Qdash SOAs with different Qdash layers shown in this paper could be important for many applications with distinct requirements under uncooled scenarios.

## 1. Introduction

Semiconductor optical amplifiers (SOAs) have been the subject of intensive research over recent decades as a key component in modern communication networks, all-optical signal processing, wavelength conversion, and many other new applications [1,2,3]. They have numerous advantages over Erbium-Doped Fiber Amplifiers (EDFAs), which are traditionally employed in optical networks, mainly in the C-band, to extend transmission distance. The gain generation in an SOA occurs directly in the active area of the semiconductor chip supplied by electrical excitation current, while in the case of an EDFA it is in the core of an optical fibre optically pumped by expensive high-power lasers. This leads to SOAs having lower costs, lower power dissipation, and smaller size, with the added advantages of both linear and nonlinear functionalities, and large-scale monolithic integration capability with silicon components and integrated circuits [4]. In addition, SOAs are implemented for a large selection of wavelength ranges depending on the materials used, including all wavelengths across the T-band (1130 nm), O-band (1300 nm), S-band (1500 nm), C-band (1550 nm), and L-band (1600 nm) [5,6]. To enhance the properties of SOAs, quantum dot/dash (QD) gain materials have been exploited with much improved performance over those using quantum wells (QWs). Performance enhancements include broader bandwidth due to the gain inhomogeneity, lower bias current and noise figure (NF), stronger nonlinearity, lesser chirp, faster response time that ensures distortion-free amplification of high-speed signals, and the ability to amplify multi-wavelength signals without crosstalk [7,8,9]. These characteristics are attributed to the electron and hole confinement in all three dimensions in the nanostructure of QD [10,11].

The study of QD SOAs and their applications have developed quickly in recent years [12,13,14,15,16,17,18,19,20,21,22,23]. QD-SOAs are excellent candidates for high-speed optical and wireless transmission applications due to their ultrafast gain dynamics and pattern effect-free amplification [2,3,12]. An all-optical method of impulse radio ultra-wideband pulse generation based on an integrated Mach-Zehnder interferometer (MZI) with QD-SOA was investigated [13]. Using QD-SOAs for ultra-fast all-optical signal processing [14], all-optical memory [15], and variable optical delays [16] was proposed. Optical gain engineering of an ultra-broadband InGaAs/AlAs solution-processed QD-SOA was investigated in [17]. In addition, InAs/InGaAs QD SOAs grown on silicon substrates were reported [18,19], achieving a wide gain bandwidth of 100 nm with a chip gain of 20 dB at 20 °C in the O-band with a central wavelength around 1310 nm. Ref. [20] investigated the high-temperature properties of InAs/InP QD-SOAs with extremely high-density QDs grown using the strain compensation technique, achieving gain of 10 dB at 50 °C, for which temperature no gain was obtained in a commercial SOA. The performance of InAs/InP QD-SOAs was investigated at even higher temperature [21] with a small-signal gain of 20 dB at 1599 nm achieved at 82 °C. A temperature-dependent shift of the peak gain wavelength at a rate of 0.78 nm/°C was observed. A record high chip gain of 35 dB at 400 mA from an InAs/InP QD SOA was reported, with a 1% bias duty cycle and 15 dB coupling loss/facet [22]; however, the chip gain calculation method they used based on calibration of the very large coupling loss and extremely small duty cycle was not convincing. In a following publication from the same group [23], a lower NF of 4.59 dB at an input of –20 dBm and 400 mA from an InAs/InP QD SOA was reported. We believe that this value of NF should actually be multiplied by a factor 2, because the equation used to calculate NF (Equation (2) in Ref. [23]) did not consider the very important polarization properties of SOA.

In all the studies above, only SOAs with a fixed number of QD layers were considered. Previously, we presented the performance of Qdash SOAs over a wide temperature range [24], and in this paper, we demonstrate a series of SOAs with different numbers of stacked Qdash layers. The overall performances of the four SOAs with three, five, eight, and twelve Qdash layers are evaluated and compared, including amplified spontaneous emission (ASE) power, spectra and bandwidth, optical gain and bandwidth, 3 dB gain saturated output power, and NF. The results demonstrate that Qdash SOAs are a promising solution for diverse applications with requirements that can be specifically met by tuning the number of Qdash layers.

## 2. Materials, SOA Structure and Measurement Methods

### 2.1. Gain Materials and SOA Structure

Figure 1a show a schematic cross-sectional diagram of an InAs/InP Qdash SOA with a single lateral mode ridge-waveguide structure. The device was grown on a 3″ (001) oriented n-type InP substrate. The detailed processes are described in [25]. Simply, Chemical Beam Epitaxy was used to grow n-type InP buffer and cladding layers followed by a 350 nm thick lattice matched InGaAsP waveguide core. The composition of InGaAsP was chosen to have a photoluminescence (PL) peak of 1.15 μm at 300 K. For the devices studied in this paper, we grew different numbers of stacked layers of Qdashes with a period of 11 nm in the waveguide core, with the top Qdash layer being 46 nm from the top of the core. Each InAs dash layer was formed by first depositing a monolayer of GaAs on the InGaAsP followed by 5 monolayers of InAs and a 25 s growth interruption to allow the InAs to diffuse on the surface and form dashes. Each dash layer was then capped with a thin InP barrier layer and finally capped with the InGaAsP barrier material. This double-cap process was used to precisely control the emission wavelength [26].

Room temperature PL for all samples gave peak wavelengths of 1564 ± 4 nm with a FWHM of 130 nm, irrespective of the number of Qdash layers. The average Qdash density in each active layer was around $1.5\times 10^{10}\;{cm}^{-2}$. A cross-sectional scanning electron microscope (SEM) image of an InAs Qdash SOA with five layers of Qdashs is shown in Figure 1b, and a top view SEM image of typical surface InAs Qdashes is shown in the inset of Figure 1b. This core structure provides both carrier and optical confinement. An upper p-type InP cladding layer, containing an etch stop for ridge fabrication and a heavily doped p-type InGaAs contact layer, was then grown with Metal Organic Chemical Vapor Deposition. Following growth, the SOAs were processed by defining ridges with a width of 2.3 µm using a combination of dry etching into the InP cladding followed by wet etching to an etch stop layer as shown in Figure 1a. Finally, a SiO_2_/Si_3_N_4_ isolation layer and a metal contact layer were deposited to form a low-resistance Ohmic contact and the devices were cleaved to a length of 1500 μm.

To reduce optical cavity effects due to the cleaved facets, the waveguide was tilted at 7° to the normal of the facet, as shown in Figure 1c. These facets were then coated with a multi-layer TiO_2_/SiO_2_ anti-reflective dielectric stack with five pairs of layers and a 1.8 µm total thickness resulting in a <10^−4^ mode reflectivity. As a consequence of the tilt, the emitted output light beams from the waveguide were at an angle of 26° normal to the cleaved surface, in air (Snell’s law).

### 2.2. Measurement Methods

Figure 2 shows a block diagram of the test setup for characterizing the ASE, gain, and NF of the SOAs. The SOA bars were mounted onto a gold-plated copper carrier on a central post to provide mechanical support. The central post was rotated 26° anticlockwise to optimize the coupling to the lensed fibers. Two needle probes were placed on the top and bottom contacts of the SOA to provide electrical connection to the chip.

To reduce temperature fluctuations, a thermoelectric cooler was embedded underneath the copper block to maintain an operating temperature in the range of 17–80 °C. An ultra-low-noise battery-powered laser diode driver controller (ILX Lightwave, Bozeman, MT, USA, Model LDC-3722) was used to provide a DC bias for the SOA. Lensed fibers were used to couple into and out of the SOAs. The output fiber was attached to a two-stage PM fiber isolator for reducing any back-reflections from the measurement setup. Two alignment stages on both sides of the SOA, with 3-axis fine alignments, were used to precisely adjust the positions of the lensed fibers for optimizing the coupling.

For calibrating the fiber coupling loss, a large-area photodetector (Thorlabs, Newton, NJ, USA, model S132C) was first used to measure the total optical power emitted from the SOA waveguide. Then, the lensed PM fiber was aligned to the SOA output waveguide and the power in the fiber was measured. In our setup, a coupling loss of 3.2 dB/facet was obtained by subtracting these two optical powers in dBm. A PM fiber optical 90/10 coupler was used with the 10% port for monitoring the coupling power using a power meter (Thorlabs, Newton, NJ, USA, models S144C and PM100D) and the 90% port for spectrum measurements using an optical spectrum analyzer (OSA) (Ando, Akasaka, Japan, model AQ6317B). 

Before measuring the SOA output spectrum using the setup shown in Figure 2, a tunable laser source (Agilent, Santa Clara, CA, USA, model 8164A) was connected to the same isolator and 90/10 coupler to measure its spectrum at each individual wavelength used in this study. The fiber-to-fiber (F-F) gain was obtained by subtracting the measured SOA output spectrum peak power after fiber alignment (at point 2 shown in Figure 2) from the measured input spectrum peak power before fiber alignment (at point 1 shown in Figure 2) at the corresponding wavelength. Therefore, the chip gain of the SOA was equal to the F-F gain plus 2 times the coupling loss. 

Considering only the signal-spontaneous beat noise [27], the noise figure is given by: (1)$$NF\left(dB\right)=\frac{2\;P_{ASE}}{G_{F-F}\;hv\;B_{0}}$$
where *P_ASE_* is the spectral density of ASE noise measured after fiber coupling, *G_F-F_* is the SOA’s *F-F* gain, *h* is the Planck’s constant, *ν* is the frequency of input wavelength, and *B_0_* is the OSA’s resolution bandwidth used in the measurements. The factor of 2 represents the polarization effect of the SOA.

## 3. Results and Discussion

### 3.1. Amplified Spontaneous Emission

Figure 3a shows the total ASE output power values versus bias current measured at 17 °C for three, five, eight, and twelve layers of Qdashes in the active region. The total ASE power values were measured by collecting all the output light using a large-area photodetector. The maximum power varied significantly for the SOAs with the different layers, as shown in the left axis of Figure 3b. The maximum power increased as the number of Qdash layers increased from three to eight layers and dropped as the number of Qdash layers increased to twelve, while the saturated bias currents in the right axis, corresponding to the maximum power, increased nearly linearly with the number of Qdash layers, as shown in the right axis of Figure 3b.

Measured ASE spectra of the different SOA devices at a bias of 300 mA and temperature of 25 °C are shown in Figure 3c. The peak power values, as indicated in the figure, increased as the number of Qdash layers increased from three to eight, and then dropped as the number of layers increased from eight to twelve, which correlates with the values of ASE total max power. Figure 3d shows the peak wavelength values of the ASE spectra, which red-shift from 1504.7 nm to 1510.4 nm, 1531.7 nm, and 1557.0 nm as the number of Qdash layers increases from three, to five, eight, and twelve, respectively. This is a consequence of the decreasing carrier density per Qdash layer as the number of layers increased since the measurements were performed at a fixed current. As the carrier density per layer increases, higher lying states of the Qdashes become occupied, resulting in an expected increase in gain at shorter wavelengths, a blueshift of the ASE peak, and a broadening of the spectrum. This is indeed observed with the 3 dB bandwidth of the ASE spectra decreasing from 90.0 nm to 64.2 nm, 42.5 nm, and 40.1 nm as the number of Qdash layers increased from three to twelve.

ASE spectra of the five-layer Qdash SOA were measured at biases of 100–450 mA at 17 °C and at temperatures of 17–80 °C at 450 mA as shown in Figure 4a,b, respectively. The ASE intensity increased and its peak shifted to shorter wavelength as the current increased. Conversely, the ASE intensity decreased and its peak shifted to longer wavelength as temperature increased. The ASE peak intensity, wavelength at peak, and 3 dB bandwidth versus current and temperature are shown in Figure 4c,d, respectively. The ASE peak intensity saturated at 400 mA for all temperatures is shown Figure 4c. The achieved ASE peak wavelength was in the range from 1515.7 to 1581.3 nm, covering the C-band and L-band. The measured ASE 3 dB bandwidth increased with both bias current and temperature, reaching 80 nm at 450 mA and 80 °C. 

Figure 4e shows the rate of change with temperature of the ASE peak intensity (left axis) and the peak wavelength and 3 dB bandwidth (right axis). Larger changes in ASE peak intensity with temperature were observed in the low-bias range, but the rate of change in peak wavelength and 3 dB BW with temperature did not vary significantly as a function of bias current. The average rate of change of ASE peak wavelength with temperature of 0.77 nm/°C is very similar to the value of 0.78 nm/°C reported in Ref. [21].

### 3.2. Gain, Gain Bandwidth, and 3 dB Saturated Output Power

Figure 5a shows the chip gain versus input optical power measured from the four SOAs with three, five, eight and twelve layers at their gain peak wavelength. From the results shown in Figure 5a, we can derive the small-signal chip gain and the 3 dB saturated output power versus the number of Qdash layers. The derived results are shown in Figure 5b,c, respectively. The small-signal chip gain of the three-layer Qdash SOA gave a relatively low value of 18.9 dB, while the 3 dB saturated output power was quite high, up to 18.2 dBm. As the number of Qdash layers increased to five, the small-signal chip gain increased sharply to 25.4 dB while the 3 dB saturated output power dropped to 15.2 dBm. When the number of Qdash layers increased to eight, the chip gain slightly increased to 26.4 dB, while the 3 dB saturated output power continued to drop at the same rate to 12.6 dBm. Finally, as the number of layers further increased to twelve, the small-signal chip gain decreased to 21.9 dB and the 3 dB saturated output power dropped to 11.7 dBm.

Chip gain was measured at different input optical powers and wavelengths for a five-layer Qdash SOA at 400 mA and 17 °C, as shown Figure 6a. The maximum small-signal chip gain of 28.2 dB was obtained at 1524 nm. The 3 dB saturated output powers for each wavelength at 300 and 400 mA and 17 and 25 °C are shown in Figure 6b. The results show the 3 dB saturated output power increased with wavelength, bias current and temperature increase. The normalized gain versus wavelength at different input optical powers, measured at 400 mA and 25 °C, is shown in Figure 6c, where it is observed that the maximum gain wavelength increased as the input power increased. The 3 dB gain bandwidth as a function of input power is shown in Figure 6d, measured at 300 and 400 mA and 17 and 25 °C. The figure shows that the gain bandwidth increased with the input optical power and bias current, but decreased as the temperature increased. The widest 3 dB gain bandwidth of 96.5 nm was obtained from the five-layer Qdash SOA at 9 dBm input power, 400 mA bias current and 17 °C. The overall gain performance achieved in this paper are better than that obtained from QW SOA [28].

### 3.3. Noise Figure and Optical Signal Noise Ratio

Due to the high gain achieved, the NF for our SOAs mainly depends on signal-spontaneous beat noise. Figure 7a shows the measured NF versus input optical power for SOAs with different numbers of Qdash layers at the corresponding wavelengths and currents. The results indicate that the NF increased as the number of Qdash layers increased, both in the small- and large-signal regions, but the increased amount was smaller in the small-signal region than in the large-signal region. For the three-layer Qdash SOA, the NF increased only 0.6 dB from 5.5 dB to 6.1 dB as the input signal power increased from −13 dBm to 7 dBm, while it increased 1.1 dB, from 7.3 dB to 8.4 dB, for the twelve-layer Qdash SOA, as shown in Figure 7b. The NFs achieved from our Qdash SOA are better than that obtained from a QW SOA reported in Ref. [28]. To compare the performance of our InAs/InP Qdash SOA with the InAs/InP Qdot SOA reported in Ref. [23], spectra of the input tunable laser source and output signal of our Qdash SOA were measured under the conditions of −20 dBm input power, 1532 nm wavelength, 300 mA bias current, and 25 °C temperature, as shown in Figure 7c. An optical signal–noise ratio (OSNR) of 39.9 dB and NF of 5.7 dB were achieved, while an OSNR of 34.3 dB and NF of 4.8 dB were reported in Ref. [23] with very similar measurement conditions; however, their NF was calculated without considering the polarization effect on the SOA compared with our measurement using PM fiber. Therefore, a correction factor of 2 is needed in their NF result. The obtained OSNR of our Qdash SOA is 5.5 dB higher than that reported in Ref. [23].

## 4. Conclusions

We investigated a series of 1500 µm long C-Band and L-band InAs/InP Qdash SOAs with different numbers of Qdash stacked layers. The overall performance including ASE, gain, bandwidth, saturated output power, and NF was compared for SOAs with three, five, eight, and twelve Qdash layers. We achieved a high chip gain of 26.4 dB at 1550 nm, with a 300 mA drive current and at 25 °C from the eight-layer Qdash SOA. A large bandwidth of 90 nm, high 3 dB gain saturated output power of 18.2 dBm, and low NF of 5.7 dB were observed for the three-layer Qdash SOA in the shorter wavelength range. A small-signal gain of 21.8 dB and a wavelength range reaching into the L-band was shown for the 12-layer Qdash SOA. The high performance from the five-layer Qdash SOA was demonstrated in detail, and the value of OSNR obtained from our SOA was 5.5 dB superior to that of the Qdot SOA reported in Ref. [23]. These results suggest that for some applications, Qdash SOAs could replace EDFAs, which would significantly reduce power consumption (from W to mW ranges), system size (from bench top to chip-scale) and cost, in addition to their attractive potential integration with Si waveguide devices.

## Figures and Tables

**Figure 1 micromachines-14-02230-f001:**
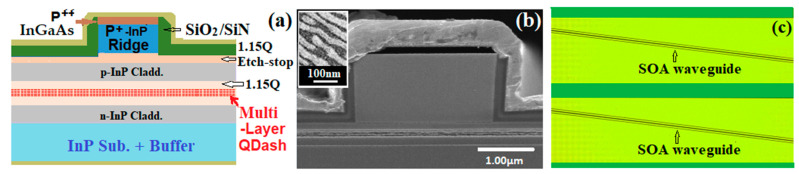
(**a**) Schematic cross-sectional diagram of an InAs/InP Qdash SOA. (**b**) Cross-sectional SEM image of a five-layer Qdash SOA. Inset: the top view of a Qdash layer. (**c**) A plan view photo of a SOA showing the waveguide tilted at 7° to the normal of the cleaved facet.

**Figure 2 micromachines-14-02230-f002:**
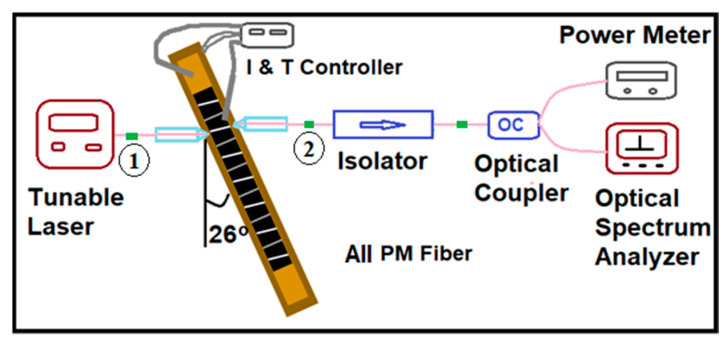
Block diagram of test setup used for characterizing the Qdash SOA. All fibers used are polarization maintaining (PM Fiber).

**Figure 3 micromachines-14-02230-f003:**
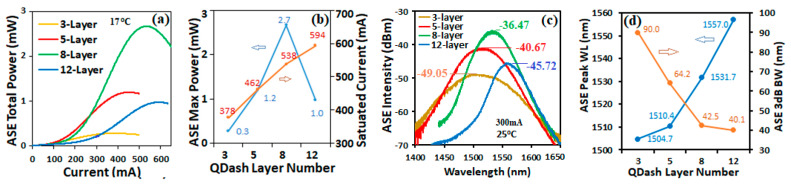
(**a**) ASE total output power of the SOAs versus bias current for the different number of Qdash layers in the active region. (**b**) The maximum ASE power (left axis) and saturated current (right axis) versus Qdash layer number. (**c**) ASE spectra at a bias current of 300 mA and 25 °C with peak powers indicated in dBm. (**d**) ASE peak wavelength (left axis) and 3 dB bandwidth (right axis) versus number of Qdash layers.

**Figure 4 micromachines-14-02230-f004:**
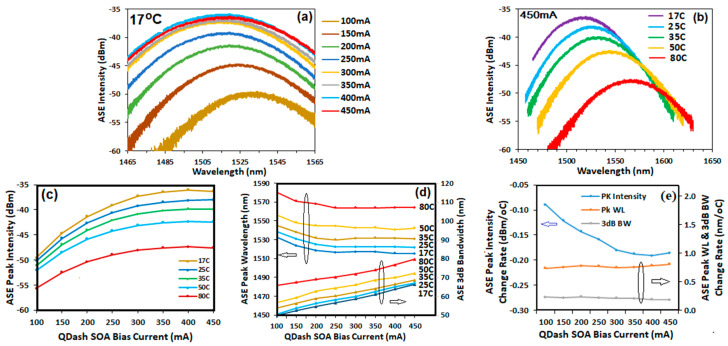
ASE spectra of the 5-layer Qdash SOA versus bias current from 100 to 450 mA at 17 °C (**a**) and versus temperature from 17–80 °C at 450 mA (**b**). ASE peak intensity (**c**), peak wavelength ((**d**) left axis) and 3 dB bandwidth ((**d**) right axis) versus bias current and temperature. (**e**) Rate of change with temperature (17–80 °C) of the ASE peak intensity ((**e**) left axis), peak wavelength and 3 dB bandwidth ((**e**) right axis).

**Figure 5 micromachines-14-02230-f005:**
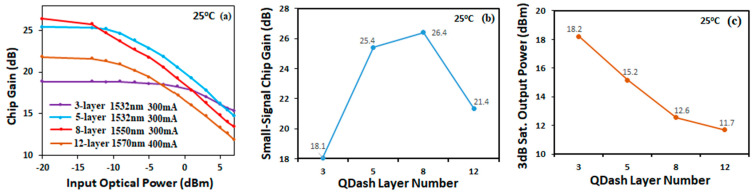
(**a**) SOA chip gain as a function of input optical power for the four different Qdash devices at 25 °C at the indicated wavelength and current. Small-signal chip gain (**b**) and 3 dB saturated output power (**c**) vs. number of Qdash layers for the indicated wavelengths and currents as shown in (**a**).

**Figure 6 micromachines-14-02230-f006:**
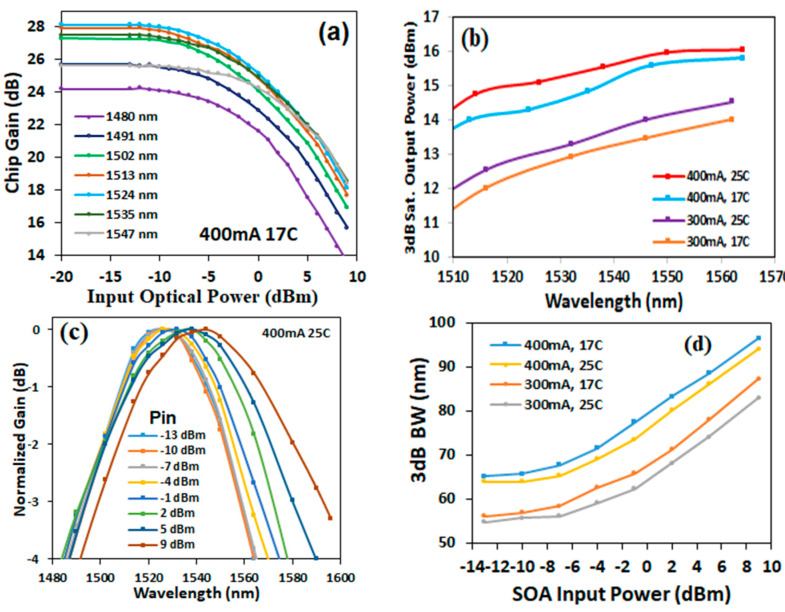
Results measured from the five-layer Qdash SOA: (**a**) Chip gain as a function of input power for different wavelengths at 400 mA and 17 °C; (**b**) 3 dB saturated output power versus wavelength at 300 and 400 mA and 17 and 25 °C; (**c**) normalised gain versus wavelength at different input powers at 400 mA and 25 °C; (**d**) 3 dB gain bandwidth as a function of input power at 300 and 400 mA and 17 and 25 °C.

**Figure 7 micromachines-14-02230-f007:**
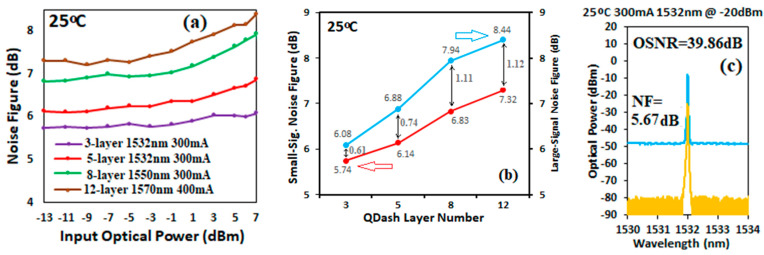
(**a**) NF as a function of input power for SOAs with different numbers of Qdash layers at 25 °C and specific measurement wavelengths and bias currents. (**b**) Small-signal NF at −13 dBm (left axis) and large-signal NF at 7 dBm (right axis) for SOAs versus Qdash layer number. (**c**) Spectra of input laser source (yellow) and output SOA (blue) measured at 1532 nm, 300 mA and 25 °C for a five-layer SOA.

## Data Availability

Data is unavailable due to restrictions.

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
