# Peer review of "Performance Investigations of InAs/InP Quantum-Dash Semiconductor Optical Amplifiers with Different Numbers of Dash Layers"

_micromachines, 2023, doi:10.3390/mi14122230_

Round 1

Reviewer 1 Report

Comments and Suggestions for Authors

The authors demonstrated QDash SOA on InP with detailed characterizations. The manuscript can be accepted after revision.

1. There are some typos. I would suggest more proofreading.

2. Why is there a gap between the InP cladding and ridge? Any crack observed on the sample?

3. It would be better if the author can specify the TiO2/SiO2 anti-reflective dielectric stack, the thickness and the stacking number.

4. What's the optimal QDash layer number? Any trade-off for different performance characteristics?

Comments on the Quality of English Language

The quality of English can be further improved. The typos should be corrected.

Reviewer 2 Report

Comments and Suggestions for Authors

This is an experimental work that showed excellent results. I would recommend the publication of this work with more analyses and explanations about their results on the following points.

1.  Line 18: The wording that reads “The longer wavelength was observed for the 12-layer Qdash SOA…” is confusing. Shouldn’t “longer wavelength” be “longer gain peak wavelength”?

2. Fig. 3 (a), why the ASE power drops for the 12-layer QDash SOA?

3. Fig. 3 (b), the 12-layer QDash SOA has a narrower ASE bandwidth, which is understandable as the current density in each QDash layer is lower. Please comment why the spectrum density of the ASE intensity is also lower? Although each layer contributes less due to the lower current density, there are more layers – theoretically they should scale in a way to make the total ASE power the same. The difference (under the same current injection) can only be from how the total ASE power is collected practically in measurement. As the structures with different layers have different confinement factors, the structure with bigger confinement factor (i.e., with more layers) should be in favor of ASE power collection in the setup shown in this work, which means more ASE power should be collected theoretically for the structure with more QDash layers.

4. Fig. 5 (a), why a higher bias current was applied to the structure with 12 -layers? And yet it small-signal gain still dropped?

5. Fig. 5 (b), the way to present the result could be misleading, as it gives the readers the impression that the structure with 5-layers offers the best saturation output power. By choosing a longer SOA length for higher gain, one cannot rule out the possibility that the structure with less layers (i.e., 3) may achieve a higher saturation output power.

6. Fig. 6 (d), it is understandable that the gain-bandwidth product increases with the bias current, but why it increases with the input signal power under the same bias? For the same bias, more minority carriers would be consumed inside the active region with increased input power (i.e., the population inversion should be less), the gain-bandwidth product should drop. If the explanation was, under smaller input power, the less consumed minority carriers would contribute to the noise, which resulted a smaller gain-bandwidth product for the signal, it contradicts against the result in Fig. 7 (a), which shows a lower NF for smaller input power.

7.  Line 268-286: Theoretically, from whatever measure or whatever sense, the Qdash is a structure in between the QD and QW structures. This work, however, showed that the NF of the QDash SOA beats that offered by either the QD or the QW SOA. Could the authors give some comments on the reasons behind this?

Round 2

Reviewer 1 Report

Comments and Suggestions for Authors

The authors have addressed all the questions. I would suggest accepting the manuscript with the current version.